# Antibody-Drug Conjugates and Targeted Treatment Strategies for Hepatocellular Carcinoma: A Drug-Delivery Perspective

**DOI:** 10.3390/molecules25122861

**Published:** 2020-06-21

**Authors:** David Dahlgren, Hans Lennernäs

**Affiliations:** Department of Pharmacy, Division of Biopharmaceutics, Uppsala University, 752 36 Uppsala, Sweden; david.dahlgren@farmaci.uu.se

**Keywords:** antibody-drug conjugates, hepatocellular carcinoma, liver cancer, drug discovery, monoclonal antibodies, bioconjugation, cytostatics

## Abstract

Increased understanding of cancer biology, pharmacology and drug delivery has provided a new framework for drug discovery and product development that relies on the unique expression of specific macromolecules (i.e., antigens) on the surface of tumour cells. This has enabled the development of anti-cancer treatments that combine the selectivity of antibodies with the efficacy of highly potent chemotherapeutic small molecules, called antibody-drug conjugates (ADCs). ADCs are composed of a cytotoxic drug covalently linked to an antibody which then selectively binds to a highly expressed antigen on a cancer cell; the conjugate is then internalized by the cell where it releases the potent cytotoxic drug and efficiently kills the tumour cell. There are, however, many challenges in the development of ADCs, mainly around optimizing the therapeutic/safety benefits. These challenges are discussed in this review; they include issues with the plasma stability and half-life of the ADC, its transport from blood into and distribution throughout the tumour compartment, cancer cell antigen expression and the ADC binding affinity to the target antigen, the cell internalization process, cleaving of the cytotoxic drug from the ADC, and the cytotoxic effect of the drug on the target cells. Finally, we present a summary of some of the experimental ADC strategies used in the treatment of hepatocellular carcinoma, from the recent literature.

## 1. Introduction

Increased understanding of cancer biology, pharmacology and drug delivery has provided a new framework for drug discovery and product development that relies on the expression of certain unique macromolecules (i.e., antigens) on the surface of tumour cells but not on non-tumour cells [1]. This knowledge, in combination with a substantial reduction in the costs associated with manufacturing biological macromolecules, has shifted the focus of tumour drug treatment from traditional parenteral chemotherapy to targeted cancer therapies using high-precision monoclonal antibodies. However, most antibody-based therapies alone result in an incomplete anti-tumour response [2]. Therefore, development of more efficient anti-cancer treatments relies on combining the selectivity of antibodies with the potency of chemotherapeutic small molecules (half maximum inhibitory concentration [IC50] in the sub-nanomolar range). These combination products have been categorised into a class of anti-cancer drugs named antibody-drug conjugates (ADCs). All ADC technologies are based on the binding of a cytotoxic drug, also called the warhead, via a linker molecule to an antibody which selectively binds to an antigen that is highly expressed on the cancer cells (or in the tumour microenvironment). Binding internalizes the ADC, whereupon the potent cytotoxic drug is released, efficiently killing the tumour cell (Figure 1).

There are multiple approaches to the development of ADC-based products, each with its own set of challenges, but the common aim is to develop an ADC with a high therapeutic/safety ratio. Generally, a successful ADC should have a relatively long terminal half-life following intravenous administration, as a low plasma clearance prolongs the time in the vascular compartment, allowing the ADC to be transported across the endothelium into the tumour tissue matrix. Release of drug from the ADC in the systemic circulation should be as low as possible; in fact, an optimal ADC design favours release only when it has been internalized into the tumour cells. Given that a suitable cancer-cell-specific antigen target has been selected, ADCs should be able to carry highly cytotoxic drugs to the vicinity of the tumour cells, reducing non-target-mediated toxic effects while increasing intracellular cancer-cell bioavailability.

Two examples of ADCs, trastuzumab emtansine and trastuzumab deruxtecan, have been clinically approved for treating human epidermal growth factor receptor 2 (HER2)-positive metastatic breast cancer [3,4]. In both, an antibody that targets HER2 (trastuzumab) is covalently linked to a potent cytotoxic agent (mertansine or deruxtecan) with IC50 values of approximately 5 nM. Mertansine induces microtubule-targeted mitotic arrest and kills tumour cells, while deruxtecan is a topoisomerase I inhibitor that induces tumour cell apoptosis [5,6]. Because cytotoxic agents like these are so potent, it is necessary that the antigen is expressed substantially more extensively on the tumour cells than on normal cells, and interpatient variability in antigen density are important factors in determining ADC efficacy [7]. Once the ADC is bound to a target cell, the onset of the anti-cancer effect is determined by the rate of internalization of the conjugate molecule and by the release rate of the active drug from the linker, which is ultimately determined by the strength of the chemical bond and the intracellular conditions [3,8]. However, the drug may also be intentionally released in the tumour microenvironment if the linker is designed to be cleaved in the extracellular tumour compartment. This requires that the drug monomer is rapidly and efficiently take up into the adjacent tumour cells, but drug exposure to normal and non-targeted tissue is also likely to be increased [9].

ADCs are classified as biological products, and their development and approval are recognized to be more complex and expensive than pharmaceutical products based on active pharmaceutical ingredients with low molecular mass (<1000 Da). Nonetheless, biologicals are becoming more important, as illustrated by the approval by the FDA of nine ADCs since the year 2000 (Table 1). Their importance is also demonstrated by the multitude of ADC targets currently explored in a preclinical and clinical setting [10]. This review will discuss some of the possibilities and limitations of the drug delivery processes that are relevant for ADCs. It will include some general theoretical considerations followed by discussions revolving around ADC transport from the blood into tumour tissue, the diffusion and convective transport of ADCs through the extracellular matrix (ECM) of tumours, the process of the antibody docking with the surface antigen, the intracellular uptake and disposition of ADCs, and the release of the active drug from the linker and its cellular fate (Figure 1). Finally, we will summarize some of the recent literature regarding experimental ADC strategies used in the treatment of hepatocellular carcinoma (HCC).

## 2. Theory

In the development and regulatory evaluation of an ADC, it is important to understand the extent of intracellular exposure (F_cell_) to the active drug substance, for both target and non-target cells, because F_cell_ is often directly related to the clinical efficacy and safety of the ADC. While the stability of the ADC and its clearance from blood are also important, the F_cell_ is primarily the result of five general processes: (i) the fraction of the active drug dose that is transported from the systemic circulation into (and throughout) the extravascular matrix (f_et_); (ii) the extravascular and extracellular metabolism of the active drug (1-E_em_); (iii) the extent of cellular uptake of the ADC (f_up_); and (iv) the rate and extent of intracellular degradation of the linker that binds the antibody to the active drug (F_m_) (Equation (1)):(1)$$F_{cell}=f_{et}\times \left(1-E_{m}\right)\times f_{up}\times F_{m}$$
f_et_ italic or not? Be consistent is affected by the size of the ADC, the inter-endothelial space, the extent of ADC diffusion and convection, and the timing of the onset of antigen docking and overall distribution in the extravascular matrix. E_m_ is determined from the metabolic and chemical stability of the ADC in the ECM in solid tumors, which can be strongly affected by the microenvironmental pathological conditions in the tumour. f_up_ is determined by the specific binding affinity between the tumor cell membrane antigen and the antibody, and its subsequent internalization rate. Finally, F_m_ is determined by the rate and extent of intracellular cleavage of the linker, releasing the active drug from the antibody (unless the drug is active while still bound to the antibody).

## 3. Vascular Endothelial Transport and Target Tissue Distribution

The generally high cytotoxicity of the active drug in ADC products requires that the amounts released during systemic circulation are kept very low [18]. Consequently, from both efficacy and safety perspectives, the transport of the ADC from the blood into the target tissue must be faster than its elimination from plasma. There is no general advice regarding the rate of these processes as all ADCs are unique, but in development it is important to understand details of the link-drug degradation process, the permeability of the cell membrane to the active drug, and the extent of protein binding, distribution volume, metabolism, and excretion of the active drug. Nonetheless, some drug release may still take place, even for antibody-drug linkers designed to be completely stable in plasma, and the ADC will be cleared from the central circulation with time by selective and unselective cellular uptake and catabolism [19].

The most commonly used antibody in ADC technology is IgG, which has a molecular mass of about 150 kDa and a diameter of about 15 nm [20]. Given its relative bulkiness, the permeation of an ADC containing this molecule across the endothelial wall is rate limiting (i.e., extravasation), and not the blood perfusion rate to tumour and normal tissues [21]. However, distribution from the vascular compartment to the extravascular tumour microenvironmental compartment is not the only transport process necessary for the ADC to reach its antigen target. Diffusion in the tumour ECM and distribution throughout the whole tumour tissue are also required, while accumulation only on/in cells in close proximity to the blood vessels should be avoided.

The question of which is the primary rate-limiting step in ADC tumour distribution, (extravasation or ECM diffusion) has been modelled using typical rates for passive antibody extravasation (blood vessel permeability to antibodies 2.8 × 10^−7^ cm/s) and diffusion (D ≈ 10 µm^2^/s, based on molecular radius), taking blood vessel surface area and density into account [22]. The results indicate that the diffusion rate inside the tumour far exceeds the permeability rate from the blood into the tumour, and typical ADC concentrations in tumours are in the range of 100- to 1000-fold lower than in plasma as a result of poor endothelial permeability [23]. One strategy for increasing endothelial transport is to reduce the size of the ADC, since passive membrane permeability is to some extent inversely related to molecular mass for this high-molecular-weight therapeutic class. However, this molecular property must be balanced, as not only membrane transport and diffusion rates increase with reduced ADC molecular size; the passive glomerular filtration rate is also expected to increase, increasing renal clearance and thus reducing total plasma ADC exposure and the terminal half-life [24]. Consequently, the distribution of an ADC into (but not necessarily throughout, as discussed below) a solid tumour is mainly determined by the vascularity, blood vessel density, and transport rate of the ADC across the endothelial wall.

An understanding of tumour pathophysiology might be advantageous for improving ADC extravasation [25]. Tumour cell proliferation and metastatic spread of the cancer require an adequate nutrient and oxygen supply, as well as removal of waste products; an inadequate blood supply can lead to tumour apoptosis. Normally, there is a rapid, local increase of vasculature blood vessels in tumour tissue, called angiogenesis. The importance of this process is highlighted by the finding that the aggressiveness of and overall prognosis for a range of cancers correlates with the levels of angiogenic factors [26]. Angiogenesis is triggered by chemical signals from cancer cells in the rapid growth phase, where both upregulation of angiogenic factors and downregulation of inhibitory factors occurs. In the formation of normal blood vessels, vascular endothelial growth factor (VEGF) is released into the surrounding tissue where it triggers endothelial cells to gradually mature into new blood vessels [27]; however, the new, rapidly formed blood vessels in cancer tissue are abnormal and leaky, with a detached pericyte and basement membrane. In contrast to normal tissue, the blood vessels in tumour tissue are heterogeneously distributed, smaller, and lack a pressure difference between the microvasculature and the interstitium. The endothelial permeability of blood vessels in tumour tissue is also increased, with subsequently more extensive extravasation of plasma proteins, macromolecules and liposomes into the interstitial space than in normal tissue. This is also true for normal inflammatory tissue, but clearance from tumour tissue is much slower if the lymphatic drainage system is also impaired. This is called the enhanced permeability-retention (EPR) effect, and it is applicable to particles and molecules with a molecular mass above the renal excretion threshold (typically >40 kDa) [25,28].

In addition to the EPR effect, it is also well established that solid tumours have a dense, pathological ECM. This can contribute to the heterogeneous tissue distribution of ADC and nano-formulations and can impede a successful anti-tumour response [29]. The combination of rapidly dividing cancer cells with the production and deposition of ECM constituents such as collagen and hyaluronic acid leads to a high solid stress in tumour ECM [30]. Solid stress arises from the mechanical forces associated with the solid phase of the tumour, and is distinct from interstitial fluid pressure, which is derived from the leaky vasculature and poor lymphatic drainage. The solid stress might collectively cause a collapse of blood vessels in tumours with a reduced tumour blood flow, which will further impede drug delivery [31]. Approaches for normalizing tumour ECM and vessels are based on remodeling the tumour microenvironment to have the properties of healthy tissue rather than completely destroying the ECM components [29,32].

Despite all the pathophysiological complexities affecting local tumour drug distribution, the EPR effect has been used in other tumour-related fields. For example, the accumulation of gallium bound to transferrin in solid tumours has been monitored in radiology for a long time [33]. Another example is Doxil^®^, a nano-PEGylated liposome formulation with a radius of about 100 nm that contains 10,000–15,000 doxorubicin molecules per liposome [34]. This PEGylated liposome product has a terminal plasma half-life of about 3 days, and the EPR effect contributes to its accumulation ratio (2.4:1) in cancer tissue compared to normal mucosa [35].

However, this EPR is quite modest and its clinical importance has been challenged, and there is a general lack of effectiveness of anti-cancer drugs in micelles or particles (or other macrostructures) intended to accumulate in cancer tissue [36]. In an extensive analysis of published reports during 2005–2015, it was found that the median amount of a drug in nanoparticle form that reached the tumour site in animals was 0.7% of the injected dose [37]. This demonstrates the difficulties associated with the directed transport of macromolecules in vivo. Part of the problem is related to the uneven development of vascular angiogenesis in cancer tissue, as this can decrease the total density, length and penetration of the blood vessels within the tumour tissue, leading to reduced blood flow, as discussed above [38]. This causes parts of the tumour tissue to be poorly perfused and consequently results in low and highly variable ADC availability, distribution and specific binding to the antigen at the tumour cells [39].

Recently, a report by Sindhwani et al. in 2020 proposed that main transport mechanism(s) for nanoparticles across endothelial cell barrier and/or into tumor cells transport is/are an active process(es) and not a passive process [40]. Different tumour vessel types and different trans-endothelial pathways need to establish the relationship between nanoparticle structure properties, such as size, shape and surface chemistry, to extravasation process(es), It is well recognized that strategies to manipulate tumour endothelium to enhance the trans-endothelial transport of antibodies are under investigations since several years [41,42].

The issues discussed above—the EPR effect and tissue distribution—are major limitations associated with the systemic administration of drugs using nano-based biotechnology for diagnosing and treating cancer. Although clinical trials with ADCs are currently ongoing, it is already obvious that clinical development has been and remains slow, and the development process is associated with a high product attrition rate. However, despite the many challenges experienced over several decades of ADC design, manufacturing and clinical development, the potential for ADCs to reach the clinics is very promising. The market approval in late 2019 of both trastuzumab deruxtecan and enfortumab vedotin-ejfv has confirmed the viability of antibody-directed delivery of cytotoxics to solid tumours [43].

## 4. The Antigen

An optimal antigen target should be expressed only on the surface of the relevant cancer cells. In reality, this ideal scenario is never met, because the genome of a cancer cell is ultimately the same as that of all the other surrounding non-tumour tissue cells. Surface targets are rarely overexpressed in cancer cells, and the approved and experimental antibodies were typically designed to target overexpressed antigens in certain types of cancer [44]. It is preferable that binding sites in a specific tissue are not saturated and that locally high concentrations can be achieved. The minimum spread of antigens per target tumour cell required to obtain selective, efficient, high-precision delivery of potent chemotherapeutic agents cannot readily be defined, as it depends also on other kinetic processes such as the internalization rate and the affinity of the ADC to the tissue [8,45]. Two examples of antigens that have been successfully targeted using ADCs are the aforementioned HER2 receptors, which are overexpressed in about 20% of human breast cancers, and CD30 receptors, which are highly expressed on some lymphoid cancers such as Hodgkin lymphoma [46,47]. The binding affinity of an ADC to its target antigen must accommodate accumulation and retention in tissue, and most ADCs have dissociation constant (K_D_) values in the range of 0.1 to 1 nM [48]. However, the binding affinity must also enable distribution of the ADC throughout the tumour, which can be hindered if the affinity is too high (further discussed in the next section) [49].

There are currently a range of ADC antigen targets being preclinically and clinically explored, thoroughly discussed by Boni et al. in 2020 [10]. All targets have in common that they are overexpressed in cancer tissue. This includes targets on cancer cells or cancer stem cells, or in tumour microenvironment such as endothelial cells and fibroblasts. Some important aspects to take into consideration in selecting a target are: (i) tissue expression and selectivity in antigen expression, (ii) target localization (surface or extracellular), (iii) the internalization and recycling of antigens situated on the cell surface, (iv) and biological function of the antigen (passive or oncogenic target).

## 5. Internalization

Typically, for an ADC to be effective, it should be internalized by the target tumour cell with an optimal drug-to-antibody ratio (DAR) of 3–4; a lower ratio may result in an insufficient drug dose, while a higher ratio may result in antibody aggregation in the ECM and/or reduced antigen affinity [50]. The internalization of ADC into the target cell should ideally be rapid and extensive as this maximizes the potential for sufficient intracellular availability and optimal subcellular compartment exposure to the active drug, as long as the ADC is not recirculated to the surface together with the antigen [23,51]. However, for many surface-bound antibodies, the dissociation rate from the antigen is slower than the rate of internalization, making recirculation less of an issue [52]. Further, the binding of an antibody to a surface antigen and its internalization can be either increased or decreased depending on whether the antibody is drug-free or drug-conjugated [53]. The binding affinity and antigen turnover/internalization rate must be balanced in the design process; if the ADC association to the antigen is too strong or internalization is too fast, tumour tissue distribution may be insufficient, whereas if antigen binding is not strong enough, the ADC will not accumulate in the tumour and/or be internalized [54,55,56]. This is called the binding site barrier, which associates the maximum antibody tissue penetration (*R*) with the interplay between tumour antigen expression (*Ag*), antibody concentration on the surface of the tumour (*Ab*), antigen turnover (*k*), antibody diffusivity (*D*), and the tissue void fraction (*v*). At steady-state, the distance from a blood vessel to which an antibody penetrates in a solid tumour is described in Equation (2) [23]:(2)$$R=\sqrt{\frac{D\times Ab}{k\times \left(\frac{Ag}{v}\right)}}$$

Equation (2) predicts the extent of tissue penetration of the ADC, which is favoured by an increased antibody dose (i.e., concentration), and reduced antigen concentrations and internalization rates. According to Equation (2), a fourfold increase in antibody dose is needed to double the penetration distance.

The (targeted and untargeted) cellular uptake mechanisms of ADCs are methodically discussed by Mahalingaiah et al. in 2019 [57]. The uptake can rely on binding to its surface antigen (e.g., clathrin-mediated endocytosis), binding to cellular receptors targeting the conserved backbone (Fc) regions of IgG (e.g., phagocytosis), as well as by non-specific endocytic mechanisms (e.g., caveolin-mediated endocytosis and macropinocytosis). Some surface antigens internalize irrespective if an ADC is bound to it or not, whereas others internalize at an increased rate when bound to a ligand [58].

Antibody internalization can be triggered, directly or indirectly, by receptor-mediated endocytosis, as demonstrated for some ADC cell-surface antigen targets such as CD33 and HER2, or by binding to proteins undergoing a normal cell surface internalization process, such as Glypican 3 (GPC3; further discussed in the last section) [59,60,61]. The type of cellular internalization mechanism may differ for different surface antigens, ranging in speed from minutes for clathrin-coated pit-mediated uptake to days for other more stable antigens that are typically identified by cell-surface-binding antibodies [23,52]. The mechanisms of internalization can also be affected by ADC binding; for example, the epidermal growth factor receptor is internalized by macropinocytosis induction following ADC binding, while in normal receptor-mediated endocytosis, macropinocytosis incorporates large amounts of the surface area [62]. At low ADC doses, the internalization process may also impact total body clearance [63]. To reach optimal therapeutic activity, the ADC internalization rate should be determined by the tumour cell endocytosis rate, rather than by systemic clearance or tissue penetration rates [64]. However, the dose, plasma clearance, and intra-tumour transport (e.g., transport across endothelial vasculature and diffusion/convection in the ECM) will contribute to the rate of availability of an ADC for internalization. Higher doses yield higher plasma concentrations and increase the likelihood of antigen saturation as well as distribution throughout the tumour prior to target cell internalization (as described in Equation (2)) [23]. The relationship between the antibody dose and the amount of ADC entering a tumour has been shown experimentally to be linear at concentrations below antigen saturation, and very high doses of antibody are typically needed to reach saturation [65].

## 6. Drug Release

The conjugated drug should ideally be released once the ADC has been internalized by the target tumour cell, as this will result in the highest intracellular active drug exposure. There are a multitude of alternative bioconjugation chemistries that aim to optimize target delivery and improve tolerability [53,66]. This intracellular release process can be catalysed by a range of mechanisms that break the antibody-drug link, such as acid-induced hydrolysis and protease cleavage of hydrazine and peptide linkers in lysosomes, or reduction of disulfide linkers in the cell cytoplasm [66]. Cleavage of the drug from the antibody is, surprisingly, not always necessary for an ADC to be effective, and there are examples of ADCs with non-cleavable linkers that retain their effect, such as trastuzumab emtansine [67]. The drug might also be released as a result of intracellular degradation of the antibody backbone in the lysosome [53,68]. However, one study showed that only ADCs with a reducible disulfide bond were able to kill neighbouring, non-antigen-expressing cancer cells in vivo, whereas the one with a stable bond only killed the antigen-positive cancer cells [69]. This additive effect is called the bystander effect, and it explains why low-antigen-expressing neighbouring cancer cells (bystanders) can also be affected by high-precision ADC treatment (Figure 1). These cells can consequently be affected by drugs that diffuse and/or are effluxed by carrier-mediated processes from the membranes of an antigen-expressing cancer cell (or leak from dying cancer cells) once the ADC has been internalized and the linker has been cleaved. Naturally, this requires that the drug can penetrate the membranes of bystander cells at a rate that exceeds its diffusion away from the tissue. In analogy to the bystander effect, non-cleavable strategies have also resulted in reduced off-target toxicity compared to cleavable strategies [70].

An ADC can be effective without being internalized by cancer cells if it is able to release the drug into the ECM upon binding and accumulating in the target tissue. These locally released free drug molecules must then enter the cancer cells to a sufficient extent, as demonstrated for some ADCs [71]. For instance, F16, a specific non-internalizing ADC that targets the glycoprotein tenascin-C excessively expressed in tumour stroma, was potent when coupled with an anthracycline cytostatic [72]. The metabolic cleavage mechanism is related to the linker, which was stable in plasma but cleaved in subendothelial ECM as a result of dying cancer cells releasing proteases. Other groups have demonstrated effective non-internalizing ADC strategies by targeting carbonic anhydrase IX, which is overexpressed on some solid tumours [73,74]. The result is ADC accumulation in antigen-positive tissue with local cleavage and cancer eradication. Clearly, this strategy requires that the extracellularly released drug molecules are rapidly taken up by the cancer cells.

## 7. The Active Drug

If substantial accumulation of ADCs in cancer cells is achieved, it is possible to use potent anti-tumour drugs with sub-nanomolar IC50 values. At the same time, high drug potency is a prerequisite, given that the total number of drug molecules attached to each ADC is limited. Consequently, the number of conjugated drugs should be as high as possible without compromising its physicochemical characteristics. Increasing the number of conjugated active drugs to more than four frequently results in ADC aggregation, increased clearance and reduced antigen affinity [44,75]. Most ADCs have a DAR of 2–4 active drugs, and the number of drugs must be balanced by the total intracellular drug exposure.

It has been estimated that a potency in the sub-nanomolar range is needed for a cytostatic agent linked to a antibody in an ADC [44]. This can be compared to the in vitro IC50 values for conventional chemotherapeutics, such as doxorubicin, which have been determined in the low micromolar range in several human cancer cell lines [76,77]. Among the potent drugs used in ADC development are derivates of dolastin 10, including monomethyl auristatin E, which was recently approved for treating Hodgkin lymphoma (Adcetris^®^) [11]. This active drug disrupts the microtubules in tumour cells from several different cell lines (HT-1080, MCF-7, and HEK-293) and kills them with a high potency (IC50 between 0.1 and 0.8 nM) [78,79]. Other examples are the DNA-damaging agents derived from synthetic pyrrolobenzodiazepine dimers, some of which exhibit IC50 values in the sub-picomolar range [80]. Cellular internalization of an ADC also enables treatment with potent cytostatic agents that normally permeate membranes poorly, with a subsequent low and variable F_cell_ (equation 1) and low toxicity when administered as a free drug [81]. The risk associated with early systemic release of such high potency, poorly permeating drugs in the central circulation is also lower than for drugs that freely enter cells. In addition, release of these poorly permeating drugs from apoptotic cells should have a lower effect on neighbouring cells, which could be seen as positive or negative.

The active drug in any ADC can be classified according to a modified parenteral biopharmaceutical classification system (mpBCS), which is based on the pharmacokinetics, pharmacodynamics and safety profile of the drug [82,83]. The system relies on the input of data describing the site of drug release, and the overall permeability of the cell membrane to the drug (Figure 2). This proposed and adapted mpBCS contains four groups comprising low and high cell membrane drug permeability, and extra- and intracellular release of the drug from the ADC. If more of the drug is released intracellularly, the ADC has the potential to be a successful development regardless of its membrane permeation (provided that the intracellular release is higher in cancer cells than in healthy cells, which is a prerequisite for any ADC product). If the chemical and metabolic release of the drug from the ADC occurs at any extracellular site (local or systemic), the development has a significantly lower chance of being successful, with the exception of active drugs that permeate membranes well and are released in the local tumour microenvironment.

## 8. Computational Molecular Docking

Computational approaches in antibody drug conjugate development for targeted tumor therapy is widely applied to investigate protein–ligand interactions and establish structure-activity relationships. This is based on the crystallographic structure of a tumor-specific antigen using X-ray crystallography and NMR experimental methods, and it is expected to improve the selectivity and anti-tumor effect resulting in a better overall in vivo performance of complex drug such as ADCs [84]. Improved computational based predictions and deeper understanding of all critical delivery steps of ADC under pathophysiological conditions is important, and it requires a strategy that optimize the design of molecular properties that maximize targeting to the tumor tissue and cancer cells and minimize drug dosage. For instance, molecular docking is then used to predict the bound conformation and binding free energy of small molecules to the target. Still, a detailed understanding of the intercalation pathway at a molecular level remains elusive. In particular, machine learning algorithms, molecular docking and molecular dynamics simulations are useful to in silico predict the optimal tumor antigen target. To improve the understanding of the molecular mechanisms in which antibody, cytotoxic drug and target antigen interact, more powerful quantum mechanics methods are necessary. In addition, to determine binding free energy of an ADC to its tumor antigen target need to have a more extensive use of umbrella sampling and steered molecular dynamics simulations. Changes in membrane dynamics upon ADC-antigen complex formation can be predicted using these methodologies which may give a hint on the tendency of the complex to undergo endocytosis. To conclude, developing a functional ADC is remarkably challenging but it can still be achieved as long as the experimental methods are complemented with appropriate computational ones. It is therefore a consensus that a better characterization of the structure-activity relationship devoted to ADCs is critical for antibody-drug conjugate research, development and future medical use. In the future there is a clear need for progress in the development of antibody-specific computational tools where their success is based on their access to more and diverse data in the public domain [85].

## 9. Hepatocellular Carcinoma

HCC is a solid liver tumour that can have different growth patterns, and either single or multiple tumours can be found [86]. Cirrhotic livers are associated primarily with multinodular HCC (multiple, scattered tumours) and diffuse HCC (multiple small nodules mimicking cirrhotic nodules), while large solid tumours, with or without small satellites, are found in non-cirrhotic livers. An HCC tumour can have distinct (expansive) or diffuse (spreading/infiltrative) borders and these are typically highly vascularized from the hepatic artery.

About 80% of all primary liver cancers are HCCs and the prevalence of HCCs is a major global health problem [87]. HCC is the second most common cause of death from cancer (following lung cancer), with a yearly incidence of about 800.000 and an annual mortality at the same level [88]. The incidence of HCC has steadily increased during the last two decades, partly as a result of the rise in obesity and type-2 diabetes, which are contributing factors in the development of HCC [89]. The development of HCC is also strongly associated with liver cirrhosis, which is caused by high, prolonged alcoholic intake, intake of aflatoxin-contaminated foods, and infection with the hepatitis B and C viruses [86]. The high prevalence of the latter two in some developing countries makes HCC about five times more common there than in developed regions. Thus, many HCC patients have two independently life-threatening liver conditions. In addition, HCC is a very heterogeneous disease, encompassing several cancer cell types, and does not appear to have the specific mutations that are often seen in other tumour types. The interactions among the tumour, the peritumoural environment, and the tumour phenotype on the efficacy of HCC treatment is largely unknown, and diagnostic biomarkers of HCC are poor. This is largely because the early tumour stages are asymptomatic and most patients are diagnosed at a late stage when curative surgery is a less likely option. This has contributed to the lack of treatment success for molecularly targeted substances in HCC [90]. In summary, the absence of an overall benefit, together with the generally poor prognosis of patients with medium and advanced HCC, results in a major medical need for new treatment strategies [91,92].

### Approved and Experimental Pharmacological Treatment Strategies

One of the options for treating intermediate stage HCC patients is transcatheter arterial chemoembolization (TACE). This is a locoregional therapy based on the intra-arterial injection of a drug delivery system composed of active drugs (commonly doxorubicin) and an embolization agent (e.g., iodinated poppy seed oil or microspheres of various polymers). However, there are several limitations associated with TACE (for example, the drugs do not penetrate sufficiently deeply into the tumour and the drug delivery system does not release the drugs at an optimal clinical rate) and the median overall survival is only increased from 16 to 19 months [93].

Sorafenib is a protein kinase inhibitor that obtained FDA approval for the treatment of advanced HCC in late 2007, and for a long time it was the only systemic agent with proven clinical efficacy in patients with unresectable HCC [94]. However, several new drugs (for example, regorafenib, cabozantinib, and ramucirumab) have recently been approved after indications that they were superior to placebo in patients unresponsive to sorafenib. There are also promising reports of immune checkpoint inhibitors, and novel drug combination strategies [90].

However, HCC is notorious for its high drug resistance, and no small molecular or antibody treatments have cured HCC to date; they only result in a modest overall increase in survival [95]. This is in contrast to many other cancers, where significant clinical outcomes have been associated with the use of new drugs in the last decade, showing that the innovative development of drugs using systemic and/or locoregional administration is challenging for HCC. New treatment strategies need to be based on the translation of preclinical studies into clinical development and eventual regulatory approval. Future innovation research and development of HCC treatments needs to recognize that HCC is not a ‘‘one drug disease”, and new treatment concepts that are characterized by individual tailored medicines need to be developed [90,96]. The application of ADCs for targeting HCC-specific antigens is therefore highly encouraged.

There are a few experimental preclinical approaches underway for ADC-based HCC treatments. One interesting target for antibody-based immunotherapies is GPC3, as it is highly expressed in more than 70% of HCCs, but is not expressed in normal adult tissues [97,98]. GPC3 is one of the six members of the mammalian glycosylphosphatidylinositol (GPI)-anchored cell surface glypican protein family, consisting of a core protein and two heparan sulfate chains. The ability of GPC3 to internalize enables the use of ADC for treating HCC [99,100,101]. The binding of a antibody to GPC3 does not trigger internalization itself, but the ADC may follow the protein when it is internalized by normal surface GPC3 recycling. The molecular mechanisms for this process have not been thoroughly described yet, but they involve binding of the hedgehog (Hh) protein to GPC3 instead of to the Hh (Patched) receptor [61]. Binding of Hh to GPC3 triggers internalization and degradation of the GPC3-Hh(-ADC) complex by a mechanism involving low-density-lipoprotein receptor-related protein-1 (LRP1) bound to GPC3 (Figure 3).

Two humanized anti-GPC3 antibody-drug conjugates (hYP7 and hYP9.1b) in the IgG format have been reported to induce antibody-dependent cell-mediated cytotoxicity and complement-dependent cytotoxicity in GPC3-positive cancer cells [102]. However, the authors concluded that naked anti-GPC3 antibodies are insufficient to cure liver cancer in mice and humans. Nonetheless, the binding affinity was excellent for GPC3-positive cancer cells, and further use of the antibodies for constructing ADCs was investigated. They went on to test hYP7, as it had the highest affinity for GPC3 and the highest stand-alone cytotoxicity, and therefore had the best potential for clinical development [103]. The ADC was composed of hYP7 coupled with duocarmycin SA or pyrrolobenzodiazepine dimer, two DNA-damaging chemotherapeutics currently used in other ADCs under investigation in clinical trials [104]. They were selected for their potency in anti-tumour response when examined in various HCC cell lines, and the ADCs were created by connecting hYP7 to one of the drugs using a dipeptide linker that was susceptible to lysosomal cleavage, thereby increasing the chance of successful intracellular drug release. Both ADCs killed GPC3-postive cancer cell lines (Hep3B, HepG2 and Huh 7) at picomolar concentrations, but the ADC based on pyrrolobenzodiazepine dimer was 5-10 times more potent than that based on duocarmycin SA. The difference between the two ADCs was not due to differences in the receptor binding affinity or the internalization rate. Instead, other plausible explanations were that the difference in potency might be related to differences in the cleavage efficiency of the linker by the lysosomal cysteine protease, cathepsin B, or the intracellular delivery of the released payload when it diffuses out from the cleavage site in the lysosome and into the nucleus where it exerts its mechanism of action [105]. On a side note, it has also been shown that bystander killing by ADCs using dipeptide linkers can be triggered by the secretion of cathepsin B from dead HCC cells, which may be advantageous if there is a diversity in GPC3 expression in the tumour [106]. Finally, it was also shown that the combination of the ADC with gemcitabine resulted in further improved anti-cancer effects both in vitro and in vivo [104].

Another promising ADC strategy relies on targeting the cell membrane-bound CD24, which is a mucin-like molecule that is overexpressed in a range of human carcinomas, including HCC [107]. This was investigated by connecting a CD24-targeting antibody to two doxorubicin molecules (G7mAb–DOX). Despite the low potency of doxorubicin (the IC50 is in the low micromolar range) compared to the active drugs in most ADCs, the ADC was still able to suppress tumour growth, decrease systemic toxicity, and prolong the survival of HCC-bearing nude mice [108]. A positive outcome for another ADC targeting CD24 has also been reported; the active drug was the previously discussed, more potent microtubule inhibitor, monomethyl auristatin E [109].

In summary it seems that there is great potential for ADCs targeting GPC3 and CD24 in HCC, given their high, selective expression in these cancer cells, and the promising results in preclinical evaluations. As the HCC tumour is highly vascularized, there are treatment opportunities for local arterial administration of ADC with a high cytotoxic load through arteries feeding well defined tumour regions, followed by embolization to increase local residence time.

## 10. Conclusions

ADCs are high precision medicines that have been designed as a vehicle for transporting a cytotoxic drug to a surface-specific target on tumour cells. This interaction is followed by internalization of the ADC into the tumour cell, and later by intracellular transport and metabolically mediated release of the active drug in specific intracellular compartments. Because of their size, chemical structure, and specificity, the pharmaceutical and clinical development of ADCs is a complex, time-consuming, and expensive process. There are several crucial physiological and pharmacological processes that need to be co-organized and optimized. The affinity between the antibody and the cellular surface antigen needs to be in the nanomolar range and the membrane internalization rate must be balanced between intracellular exposure and the tumour tissue distribution kinetic processes. The cleavage of the linker should be rapid following internalization and the cytotoxic effects of the drug should occur in the nanomolar range. Despite the complexities of ADC delivery, this is a promising approach for treatment of a range of cancers with currently suboptimal treatment options. HCC is a very diverse disease with multiple cancer cell types, and the limited overall benefits of the current treatments results in a large unmet medical need. Two potential cell-surface protein targets are GPC3 and CD24, as they are highly expressed in HCC. ADCs containing GPC3 or CD24 antibodies may provide a more accurate and effective way of treating liver cancer, especially when systemic therapy is combined with locoregional trans-arterial administration directly into the vicinity of the HCC.

## Figures and Tables

**Figure 1 molecules-25-02861-f001:**
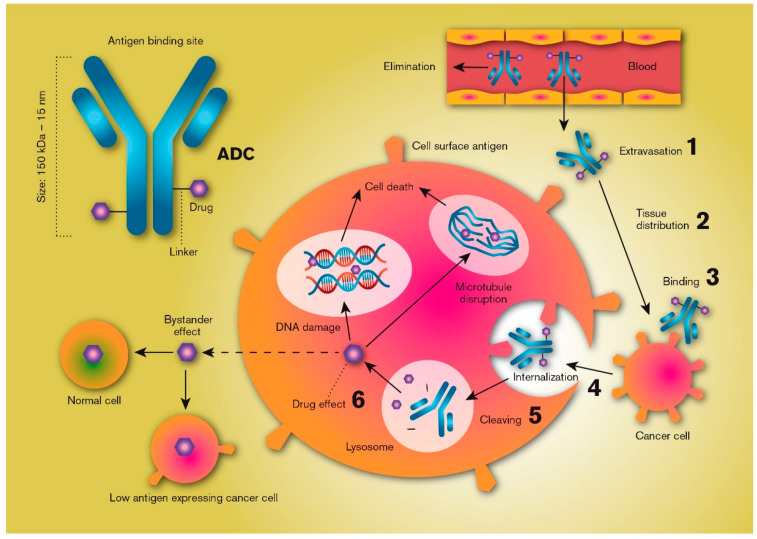
Schematic illustration of the key processes that determine the local distribution and effects of an antibody-drug conjugate (ADC). ADCs are products based on selective targeting, efficient internalization, and site-specific cleavage in the tumour cell resulting in the high intracellular availability of very potent chemotherapeutics. Critical features include plasma/systemic stability, tumour tissue diffusion/distribution, target selection, cell uptake characteristics, linker chemistry and cleaving mechanisms, and antibody-to-drug molecule ratio.

**Figure 2 molecules-25-02861-f002:**
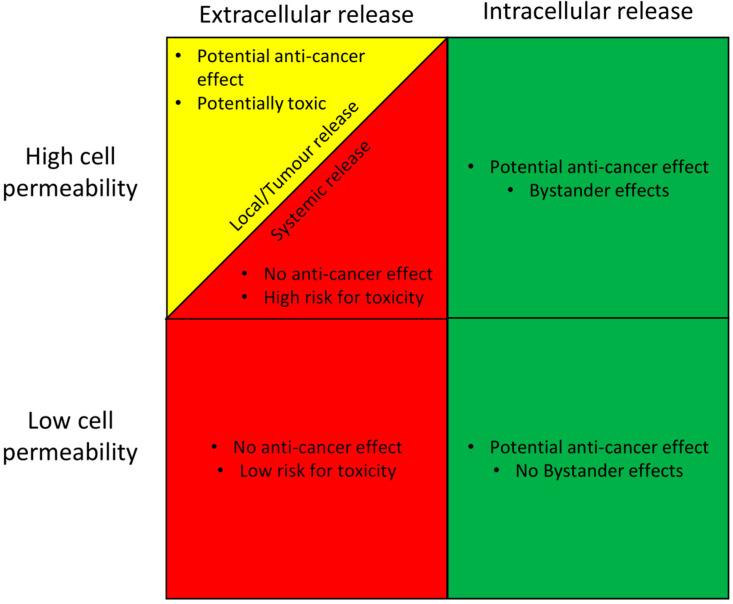
A proposed classification system for the clinical suitability of an antibody-drug conjugate (ADC) based on its site of active drug release, and the rate of cell membrane permeability of the active drug. ADC suitability is graded as good (green), intermediate (yellow), or poor (red), based on its prospect of having an anti-cancer effect, and also of causing side effects. The proposed scheme is based on the well-established Biopharmaceutical classification system [82].

**Figure 3 molecules-25-02861-f003:**
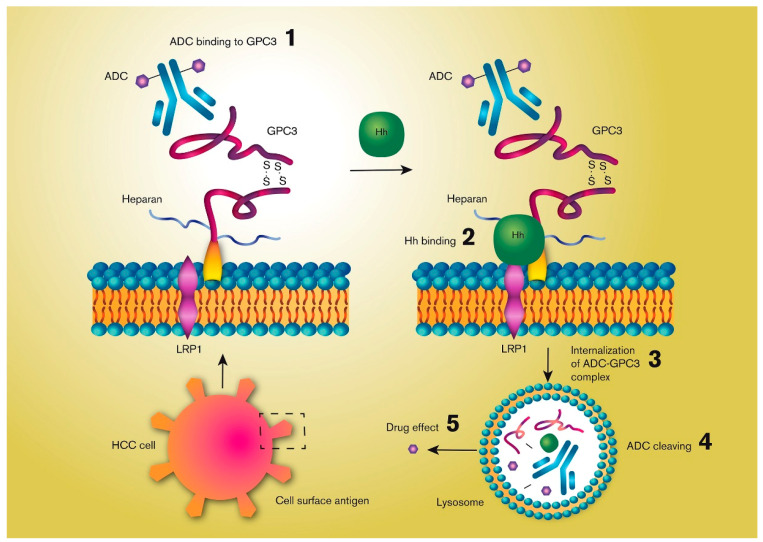
Simplified illustration of the mechanisms by which an antibody-drug conjugate (ADC) may be internalized by a hepatocellular cancer (HCC) cell by targeting the cell surface membrane-bound Glypican-3 (GPC3), a heparan sulfate proteoglycan protein. GPC3 internalization is triggered by its binding to the hedgehog (Hh) polypeptide ligand by a mechanism involving low-density-lipoprotein receptor-related protein-1 (LRP1). An ADC bound to GPC3 can be degraded together with GPC3 following its internalization, which triggers release of the active drug.

**Table 1 molecules-25-02861-t001:** Summary of the currently FDA approved antibody-drug conjugates (ADC), including their cellular targets, drug and drug mechanisms, and indication.

ADC	Antigen Target	Drug	Drug Mechanism	Indication
Brentuximab vedotin [11]	CD30	Auristatin	Microtubule disruptor	Hodgkin lymphoma
Gemtuzumab ozogamicin [12]	CD33	Calicheamicin	DNA damage	Acute myeloid leukemia
Inotuzumab ozogamicin [13]	CD22	Calicheamicin	DNA damage	Acute lymphoblastic leukemia
Moxetumomab pasudotox [14]	CD22	PE38	Apoptosis induction	Hairy cell leukemia
Polatuzumab vedotin [15]	CD79b	Auristatin	Microtubule disruptor	B-cell lymphoma
Enfortumab vedotin [16]	Nectin-4	Auristatin	Microtubule disruptor	Bladder Cancer
Trastuzumab deruxtecan [4]	HER2	Deruxtecan	Topoisomerase I inhibitor	HER2 positive breast cancer
Trastuzumab emtansine [3]	HER2	Maytansine	Microtubule disruptor	HER2 positive breast cancer
Sacituzumab govitecan [17]	Trop-2	SN38	Topoisomerase inhibitor	Triple-negative breast cancer

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
