# Peer review of "Antibody-Drug Conjugates and Targeted Treatment Strategies for Hepatocellular Carcinoma: A Drug-Delivery Perspective"

_molecules, 2020, doi:10.3390/molecules25122861_

Round 1
Reviewer 1 Report
The review manuscript by Dahlgren and Lennernas is focused on the ADCs and their application in hepatocellular carcinoma. The manuscript is well-written and contains suficiently detailed information. I have few suggestions that may help to further improve the authors work:
- The authors should put a little bit more attention to the mechanism of ADCs action, especially antigen properties as well as internalization and intracellular trafficking. Which antigens are currently tested/selected as ADC targets and why? What is the mechanism of ADCs internalization? What is the relation between antibody affinity and their uptake? Which endocytic mechanisms do ADC employ? Do they enter the cells via mechanism dictated solely by antigen or they can modify the antigen uptake when bound to it? What is the impact of ADCs size on their endocytosis? What is the impact of ADCs valency on their uptake?
- There are several non-conventional drug vehicles that are proposed instead of antibodies for ADCs, eg. natural growth factors like FGFs or EGFs. Their applicapility for targetted cancer therapy instead of antibodies should be discussed (e.g. in the outlook section).
Reviewer 2 Report
The authors details the summary on antibody drug conjugate and various factors on which drug antibody therapeutics reply upon including target tissue distribution, the antigen or receptor they bind, the internalization into cells, drug release and finally some insight into the application of antibody based therapeutics on hepatocellular carcinoma. All together the article has relevant information and could be accepted for publication after some minor correction as detailed below.
- The numbering for the section “Hepatocellular carcinoma” and “Conclusion” both have been marked as 8. It needs to be corrected as 8 and 9 respectively.
- The subheading on line 372 “Approved and experimental pharmacological treatment strategies” should be 8.1 and not 7.1.
Reviewer 3 Report
Thanks for providing such a good review on antibody-drug conjugates (ADCs) related to liver cancer. Since the review has covered complete aspects on the topic, I have no particular comments on the review. It would be better to provide one table to summarize current clinical approved ADCs as well as the ADCs under human clinical trials. After that, I recommend acceptance of the manuscript for publication.
